# Effect of an Educational Intervention on Pupil’s Knowledge, Attitudes, Perceptions, and Behavior on Air Pollution in Public Schools in Pristina

**DOI:** 10.3390/ejihpe15050069

**Published:** 2025-05-02

**Authors:** Zana Shabani Isenaj, Hanns Moshammer, Merita Berisha, Lisbeth Weitensfelder

**Affiliations:** 1Medical Faculty, University of Hasan Pristina, George Bush 31, 10000 Pristina, Kosovo; merita.berisha@uni-pr.edu; 2Department of Environmental Health, ZPH, Medical University of Vienna, 1090 Vienna, Austria; lisbeth.weitensfelder@meduniwien.ac.at; 3Karakalpakstan Medical Institute, Nukus 230100, Uzbekistan

**Keywords:** air pollution, intervention, knowledge, attitudes, perception and behavior, pupils, schools, environmental health literacy

## Abstract

This interventional study aimed to assess the effectiveness of a school-based environmental education program on improving knowledge, attitudes, perceptions, and behavior related to air pollution among pupils in low-middle schools in Pristina, Kosovo. Air pollution is a pressing issue in Kosovo, particularly in urban areas, making it essential to raise awareness from an early age. As one of the first initiatives of its kind in the country, this study offers valuable insights into the impact of educational interventions on students’ understanding of environmental issues. The study involved an intervention group of fifth to ninth grade students who participated in a structured environmental education program, with data collected through pre-test, post-test, and follow-up assessment. We used a quantitative questionnaire with four sections—demographics, knowledge, perceptions, attitudes, and behavior. The findings revealed a significant improvement in knowledge and perceptions about air pollution among students in the intervention group, highlighting the crucial role of education in raising environmental awareness. However, the intervention had limited impact on changing attitudes and no significant effect on pro-environmental behavior, echoing challenges found in previous studies. Parental education, particularly maternal education, was found to play a substantial role in shaping attitudes, while gender and parental education positively influenced perceptions. The study also identified a negative association between higher grade levels and both knowledge and perception scores. Despite its success in enhancing knowledge, the short intervention period and challenges in participant engagement limited the program’s ability to drive long-term behavioral change. These findings emphasize the need for more sustained and comprehensive interventions to address the complex relationship between knowledge, attitudes, and environmental behaviors.

## 1. Introduction

Air pollution is a worldwide environmental issue that significantly increases rates of premature death and illness, particularly from cardiovascular and respiratory diseases and cancers, especially in urban settings ([17]). Additionally, air pollution increases the risk of stillbirth, miscarriage, and neurological conditions such as cognitive impairment and dementia ([54]). Exposure to ambient air pollution, primarily from fine particulate matter (PM2.5), has been linked to both short-term and chronic health problems, affecting all organs, exacerbating existing health conditions ([42]). According to the World Health Organization (WHO), in 2019, air pollution was responsible for 6.7 million deaths, becoming the second leading cause of non-communicable diseases (NCDs) globally after tobacco ([54]). While everyone is impacted by air pollution, those with pre-existing medical issues, the elderly, children, and pregnant women are more susceptible ([54]). People from low-socioeconomic groups in large parts of Europe are more likely to live next to major roads and industrial areas, thus facing a high exposure to air pollution ([16]). Furthermore, less developed countries in the Eastern part of Europe use low-quality solid fuels such as coal and wood for the majority of their energy production, leading to high levels of exposure in the low-income population to PM2.5 levels ([16]). Kosovo, a Western Balkan country, uses coal for 95 percent of its power generation, primarily lignite, a highly polluting fuel, responsible for approximately 760 premature deaths every year ([4]; [52]). Given the magnitude and consequences of this worldwide public health threat, numerous interventions have been developed to reduce air pollution exposure and its adverse health effects ([9]). Among these, significant efforts have been directed towards raising community awareness about the associated health impacts of air pollution and its primary sources. Enhancing public literacy on the sources and the impacts of air pollution and interventions to reduce air pollution is a crucial approach to tackle this problem ([53]). Among strategies that could help reduce the effect of air pollution are awareness raising interventions, risk communication strategies, and environmental education ([37]; [33]; [36]). Understanding and responding to outdoor air pollution is crucial for urban health. While research often focuses on the environmental knowledge and behaviors of college and high school students, there is limited research on elementary school children’s environmental awareness ([31]). Environmental education is essential for promoting sustainable environmental conservation and can drive behavioral changes, reducing personal exposure and emissions and promoting eco-friendly transportation methods like walking, cycling, and public transport ([28]). The aim of environmental interventions at the primary school level is to nurture future citizens who can make informed decisions and contribute positively to society. Additionally, educating young children about their right to live in a clean environment and the significance of environmental health for their future is a crucial strategy for combating these risk factors ([27]). Therefore, it is vital to implement effective strategies that make them more knowledgeable and concerned as this ultimately leads to responsible behavior. Despite the widespread recognition of environmental education as a vital tool in addressing global challenges such as air pollution, there is a notable gap in the research, especially in low- and middle-income countries. Most existing studies have focused on environmental education in high-income countries, with little emphasis on its effectiveness in regions facing significant environmental health challenges ([19]). [20] ([20]) highlighted that a scarcity of research evaluating environmental education programs is a particular challenge in less-developed regions. This research gap is particularly evident in the context of air pollution education, especially in countries like Kosovo, where the impact of air pollution is most pronounced. While environmental education has been integrated into school curricula in various countries, there is limited evidence on its effectiveness in increasing knowledge, shifting attitudes, and changing behaviors related to outdoor air pollution, particularly among younger children ([31]). This deficiency in environmental knowledge may be attributed to factors such as inadequate educational strategies or interventions and a lack of evidence-based approaches in environmental education. Although there are numerous global publications on indoor air pollution and the impact of such interventions on knowledge, perceptions, and behavior within school settings, there is significantly little research focusing on outdoor air pollution interventions ([48]; [2]). Kosovo represents an important case study in this regard, given the severity of its air pollution problem, particularly in Pristina, where pollution levels are particularly high during the winter months. The combustion of solid fuels, outdated thermal power plants, industrial emissions, and vehicle exhaust contribute to elevated particulate matter levels, worsening respiratory and cardiovascular diseases. Despite the importance of environmental education, there have been no substantial efforts to evaluate the effectiveness of national strategies, such as the development of school environmental projects, in increasing students’ knowledge and fostering positive behavioral changes related to air pollution. Specifically, there is a lack of studies focusing on the impact of outdoor air pollution education interventions on students’ attitudes, behaviors, and long-term knowledge retention, particularly at the primary school level.

This study is one of the first to evaluate an environmental education program targeting outdoor air pollution in Kosovo, addressing a significant gap in the literature. By focusing on this under-researched area, the research not only contributes to the body of knowledge on environmental education in low-income countries but also provides valuable insights into how such interventions can be tailored to the unique challenges faced by regions like Kosovo. The findings will offer a deeper understanding of the effectiveness of school-based interventions in shaping children and adolescents’ knowledge, attitudes, perceptions, and behavior towards outdoor air pollution and its associated health risks. The primary aim of this study is to evaluate the long-term impact of an educational intervention on outdoor air pollution, focusing on knowledge retention, changes in attitudes, perceptions, and behaviors among elementary school students. Specifically, the study will compare data from pre-test, post-test, and follow-up assessments to assess both short-term and long-term effects. Moreover, the study will utilize regression analysis to explore potential relationships between various factors that may influence environmental knowledge, behaviors, and attitudes, independent of the educational intervention itself. This approach will address the research questions by examining the effects of the educational intervention on environmental knowledge, attitudes, perceptions, and behavior, as well as the role of socio-demographic variables in shaping these outcomes. Ultimately, this research will contribute to the broader field of environmental education by demonstrating how such programs can foster short-term and long-term positive changes. It aims to cultivate a generation of environmentally conscious individuals who are better equipped to advocate for and implement solutions to reduce air pollution, especially in regions like Pristina, where outdoor air pollution is a significant public health concern.

## 2. Materials and Methods

### 2.1. Study Design

This study employed an intervention-control design, conducted from 2023 to 2024 in the region of Pristina, Kosovo. A total of eight schools participated—four intervention schools and four control schools. Initial analyses revealed variations in knowledge outcomes, leading to adjustments in the instruments to improve their sensitivity. In 2024, the focus shifted to comparing knowledge, attitudes, perception, and behavior outcomes between intervention and control groups to assess the impact of the educational intervention.

### 2.2. Participants

The study involved children from low–middle schools in Pristina, Kosovo, specifically from 5th to 9th grades. Eight schools were selected, with four schools randomly assigned to the intervention group and four to the control group (Figure 1: Flow Diagram).

### 2.3. Description of Intervention

The educational intervention comprised three key components: 1. Sessions: The principal investigator conducted interactive educational sessions with pupils in the intervention group. These sessions covered the following: Air pollutants: Key pollutants, their sources, and how they affect air quality in Kosovo. Health impacts: The connection between air pollution and specific health issues, emphasizing respiratory and cardiovascular conditions. Prevention strategies: Practical measures to reduce exposure, such as staying indoors during high-pollution periods and promoting green behaviors like using public transportation and planting trees. The intervention was developed based on insights from the baseline survey, which identified specific knowledge gaps and allowed for the tailoring of content to target areas where understanding was limited. We also considered the existing curricula on environmental education and aligned the sessions with these frameworks. Additionally, the intervention was reviewed by the National Institute of Public Health of Kosovo to ensure its relevance and effectiveness. While we did not directly test activities with the target audience beforehand, future studies could benefit from engaging children in co-designing activities to ensure the most effective communication strategies. 2. Short Messages: Pupils in the intervention group received concise, actionable reminders about air quality and protective behaviors. Examples of these messages included: “Check the air quality before heading outdoors—protect your lungs!”; “Reduce air pollution by walking, biking, or taking the bus. Small actions lead to big changes!” These messages were designed to reinforce the key themes discussed during the educational sessions, ensuring ongoing engagement. 3. An educational booklet was distributed to the intervention group, providing a user-friendly guide on air pollution and prevention. Key sections included the following: Definitions and examples of common air pollutants; illustrated health impacts of exposure to polluted air; preventative tips relatable to children, such as choosing indoor activities on high-pollution days. The booklet aimed to serve as a lasting resource for pupils, encouraging them to share their knowledge with family members and peers. Schools in the control group did not receive any of the intervention components; however, we plan to distribute the educational booklets to them after the study concludes. Researchers interested in replicating these sessions may contact the authors for access to the materials. The sessions were interactive, featuring group discussions, visuals, and examples relevant to the students’ daily lives. The content was consistent across all grade levels, as it was aligned with the curricula prior to the intervention. Each session lasted approximately 45 min and was conducted over a period of 3 weeks for each grade level. Sessions were held with class groups of about 25–30 students, ensuring a manageable group size for effective engagement and discussion.

### 2.4. Instrument and Construction of Scores

Since some of the knowledge questions did display some inconsistencies in the previous study, a Rasch analysis was performed to exclude the least discriminating questions. To ensure the fairness of comparing individuals based on a single overall knowledge score, we used the Rasch model ([38]), which assumes the probability of answering a question correctly depends solely on the individual’s ability and the item’s difficulty. If other factors systematically influence item responses, the Rasch model is violated, and the overall score cannot be considered an accurate measure of knowledge ([26]). Given that our study focuses on knowledge assessment, we tested whether the Rasch model holds for our dataset. These calculations are based on the pre-test dataset (prior to any interventions), as the primary objective was to determine whether the scale score was generally a fair measure, regardless of whether participants were in the training or non-training group. Since the model applies only to dichotomous data, we combined “don’t know” and incorrect answers into a single “incorrect” category. We applied a differential item functioning with split criterion based on the median, where the difficulty ratio between items should remain consistent across high- and low-performing groups. Our initial analysis found that five of the thirteen items violated the Rasch model assumptions. Although further analysis and model refinement were beyond the scope of this study, we opted to exclude the most problematic items to enhance the fairness of the remaining knowledge score. The results from our Rasch model analysis can be found in the Appendix A.

For the other scores, the same questions were used as in the previous study. The questions used are presented in Appendix B. The knowledge questions in the Appendix B are not only closed questions (true/false) but also open-ended questions.

As explained, the knowledge answers were recorded as either correct or wrong, the latter also including the “don’t know” response, coded each as 1 or 0, and the knowledge score was calculated by adding up the results of all 8 answers. Questions about attitude and perception usually allowed for 5 answers ranging between “not at all” and “completely” or “never” and “always” and were coded 0 to 4 each. The 3 behavior questions allowed 4 answers (Never/Rare/Sometimes/Always) that were coded 0 to 3. Scores were always constructed by adding all answers per category. The question with multiple options was converted into a 0–1 code where 0 represents “no” or “don’t know” (false responses) and 1 represents “yes” or “true” (affirmative responses), even if multiple answers are given.

### 2.5. Data Collection

Data were collected in two phases: baseline in 2023 and post-test and follow-up in 2024. The baseline survey helped refine the assessment instrument, and the post-test measured the impact of the intervention. Both groups completed the same questionnaire at both time points to assess the changes in the investigated outcomes.

### 2.6. Statistical Analysis

A spreadsheet was used to enter the data, and STATA Version 17 was used for analysis. Descriptive statistics were calculated, and to examine differences between intervention and control groups at both post-intervention and follow-up time points, *t*-tests and ordered logistic regressions were performed. While the *t*-tests allowed us to compare intervention with the control group, the regression analysis of the ordinal outcome data allowed us to study the effect of additional individual determinants like parents’ education, gender, age, and health status. Because of the smaller dataset at follow-up, the analysis was restricted to simple *t*-tests. Statistical significance was set at a *p*-value of less than 0.05.

### 2.7. Ethical Approval

Ethical approval was obtained from the Medical Faculty of University of Hasan Pristina, with subsequent approvals from the directorate of the education in the municipality of Pristina and voluntary agreement of each participating school. Verbal consent was accepted as agreement to participate in the survey, and students were advised that it was anonymous and that they might discontinue participation at any moment.

## 3. Results

### 3.1. Descriptive Statistics

For all 2396 participants (1169 in the intervention group and 1227 in the control group who completed the post-test) and 1059 participants (510 in the intervention group and 541 in the control group), their follow-ups underwent final analysis. The results of the post-test and follow-up are summarized in Table 1.

### 3.2. Regression Analysis

The knowledge score was significantly influenced by the intervention and grade. The intervention had a strong positive effect (OR = 26.83, 95% CI [22.12, 32.54], *p* < 0.001), substantially improving knowledge scores post-intervention. In contrast, higher grades were associated with lower knowledge scores (OR = 0.78, 95% CI [0.73, 0.82], *p* < 0.001), indicating that as grade levels increased, knowledge scores tended to decrease slightly. Gender showed no significant impact on knowledge (OR = 0.91, 95% CI [0.78, 1.05], *p* = 0.075; Figure 2).

Our analysis reveals that the attitude score is significantly influenced by grade, mother’s education, and gender. Mother’s education had the strongest positive impact (OR = 1.20, 95% CI [1.09, 1.31], *p* < 0.001), indicating that higher maternal education levels are associated with improved scores. Grade level also showed a positive effect (OR = 1.09, 95% CI [1.03, 1.15], *p* = 0.003), suggesting a slight increase in scores with advancing grade. Gender had a significant effect (OR = 0.85, 95% CI [0.74, 0.98], *p* = 0.023), highlighting disparities between genders, with boys scoring significantly lower than girls. The intervention had a positive but marginal effect (OR = 1.15, 95% CI [0.99, 1.32], *p* = 0.059; Figure 3).

Our analysis indicates that the perception score is significantly influenced by the intervention, grade, and gender. The intervention had a strong positive effect (OR = 1.92, 95% CI [1.66, 2.21], *p* < 0.001), greatly improving perception scores. In contrast, higher grades were associated with a slight decrease (OR = 0.92, 95% CI [0.87, 0.97], *p* = 0.002), and gender showed a notable effect (OR = 0.60, 95% CI [0.52, 0.70], *p* < 0.001), revealing disparities between genders, with girls scoring higher than boys. Father’s education had a marginal impact (OR = 0.91, 95% CI [0.82, 1.00], *p* = 0.052; Figure 4).

Our analysis shows that the intervention has a significant negative effect on behavior scores (OR = 0.68, 95% CI [0.59, 0.78], *p* = 0.008), indicating it lowers scores. Higher grades also exhibit a marginally non-significant negative effect (OR = 0.95, 95% CI [0.90, 1.00], *p* = 0.067). In contrast, mother’s education significantly negatively impacts behavior scores (OR = 0.86, 95% CI [0.79, 0.95], *p* < 0.001). Additionally, gender disparities are evident, with a significant negative effect (OR = 0.80, 95% CI [0.69, 0.92], *p* = 0.002). Overall, both the intervention and mother’s education have substantial negative effects, while grade effects are marginally non-significant, and gender also shows a significant impact. Similar to attitudes and perception, boys scored significantly lower than girls in behavior, indicating a consistent trend across all measured aspects. Figure 5).

### 3.3. Follow-Up Analysis

In the follow-up analysis of 541 control and 510 intervention participants, the intervention group demonstrated significant improvements in knowledge, attitudes, and perceptions. Knowledge scores (Table 2) averaged 5.41 in the intervention group, compared to 3.03 in the control. Attitude scores (Table 3) were also higher in the intervention group (8.18 versus 7.72), as were perception scores (Table 4) (22.69 versus 21.37). However, behavior scores (Table 5) were unexpectedly higher in the control group, with an average of 5.33 compared to 4.65 in the intervention group. Overall, the intervention positively impacted knowledge, attitudes, and perceptions, while behavior outcomes contrasted with this trend.

## 4. Discussion

This interventional study was designed to show whether an innovative school-based environmental health intervention effectively contributes to a wide range of outcome parameters, such as knowledge, attitudes, perceptions, and behavior of pupils from low–middle schools in Pristina towards air pollution, a significant environmental issue in Pristina. Kosovo. To our understanding, this is one of the first investigations conducted in Kosovo; however, it is also unique with regards to interventional studies towards outdoor air pollution and aiming to improve awareness towards this emerging issue.

Knowledge—Overall, the findings of our study showed that the intervention was effective in improving the general knowledge and perceptions of students towards air pollution. Knowledge is considered as a basic factor in the process of behavior change and is key to improving urban air quality ([21]). The knowledge score was considerably higher among the intervention groups in our study. A study conducted in Mexico to evaluate the effectiveness of an environmental education program for pupils showed significant improvement in knowledge by 16.3% among students in the intervention arm ([40]). Furthermore, evaluation studies of environmental interventions showed significant improvement in students’ knowledge of environmental issues ([41]; [50]). This highlights the importance of environmental education interventions to improve students’ ecological knowledge acquisition. The improvement in knowledge and perceptions towards air pollution through the intervention confirms the findings of previous studies ([46]; [45]; [7]). Thus, it highlights that efforts targeted at enhancing awareness and environmental knowledge with high quality interventions are a potential solution to halt environmental issues such as air pollution. The ordered regression modelling identified the intervention and grade level as having a significant association with the knowledge score. Consistent with our previous study results, the analysis revealed an inverse relationship between grade level and overall knowledge, with higher-grade students exhibiting poorer knowledge ([44]). This aligns with the earlier research, which also found that higher-grade students tend to demonstrate lower levels of environmental knowledge ([10]). The decline in environmental knowledge with increasing grade could be attributed to a lack of age-appropriate activities or ineffective implementation in the education system. It was concluded that environmental training needs to be organized based on students’ age groups for better effectiveness ([3]). However, other studies report grade-dependent disparities in environmental knowledge, with knowledge levels often increasing with grade, highlighting inconsistencies with the findings of our study ([44]).

Attitudes—The intervention did not show any significant improvements in attitudes. These results were in line with previous research showing limited effect on transforming student’s attitudes ([14]; [39]; [32]). However, contradictory results were found in other studies. For instance, [34] ([34]) found significant improvement in attitudes in private eco-schools in Turkey.

Our findings show a consistent effect of grade level on attitude scores, with an increase in positive attitudes as grade level increases. In further investigating attitudes, the analysis reveals that parental educational attainment plays a significant role in shaping attitude scores, corroborating previous findings from other studies ([18]; [29]). Notably, maternal education emerged as a stronger predictor of positive attitudes compared to paternal education, consistent with the patterns observed in our baseline assessment ([44]). The substantial coefficient for maternal education underscores the pronounced impact that increasing levels of maternal education have on enhancing attitude scores. In contrast, paternal education, while still positively associated with attitude scores, exhibited a less pronounced effect. This differential impact suggests that maternal education may have a more direct or robust influence on attitudes, potentially due to maternal roles in early childhood socialization and education ([5]). Gender seems to affect attitudes, indicating differences between male and female participants, which is consistent with the baseline findings ([44]). Similar to our findings of girls expressing higher pro-environmental attitudes, [51] ([51]) found that women in their study were more likely than men to express pro-environmental attitudes, often explained by their caregiving responsibilities in the home and society. Furthermore, [55] ([55]) evidence the fact that women are more likely than men to exhibit pro-environmental attitudes ([55]).

Perception—The analysis reveals that the intervention significantly enhanced risk perception scores, while gender and parental education are negatively associated with improved perception scores. A previous study by [30] ([30]) revealed that targeted interventions influenced the way people perceived air pollution and reconsidered their protective actions towards it. In contrast, higher grade levels are negatively associated with perception scores, indicating that students in higher grades may have less favorable perceptions. Studies by [47] ([47]) and [1] ([1]) found that younger students have more positive environmental perceptions, knowledge, and attitudes than older ones. This may be due to generational differences and changes in socio-economic status with age. Younger individuals are typically more idealistic and willing to advocate for the environment, while older individuals tend to maintain the status quo. Regarding gender disparities, girls outperform boys in perception.

Behavior—Furthermore, one of the outcomes of interest of the current study was also changes in sustainable behavior in the intervention group. Our analysis revealed that similarly to the attitudes scores, the intervention did not show any positive effect on the behavior, which is in line with a previous study conducted by Pauw and Van Petegem ([14]; [35]). The study reported that the intervention itself had a negative effect on behavior outcomes. Mother’s education and gender also seem to play a role in the way pupils behave towards the environment, confirming the interactions from the baseline assessment. This result aligns with the existing literature that highlights the influential role of maternal education in child development and behavioral outcomes ([5]). Likewise, with attitudes, differences have been identified among male and female with regards to pro-environmental behaviors. The research evidence with regards to this, however, has been somehow inconsistent throughout the years. For instance, most research has indicated that women tend to act more environmentally friendly, but some studies have found no gender differences or that men exhibit more environmentally unfriendly behavior. For instance, women reported adopting ecologically responsible habits at a considerably higher rate than men, according to a meta-analysis of studies conducted in 14 different countries ([55]). In contrast, [6] ([6]) found no significant difference between males and females, whereas [15] ([15]) indicated that men had stronger pro-environmental actions than women ([6]; [15]). Both our baseline data and other studies, such as a study by ([22]), show that parental education, especially that of mothers, has a considerable influence on children’s behavior. The association was negative in every instance, indicating that the scores on the variables tended to decline as parental education levels rose. The trend changes though when looking at behaviors associated with consumption. In general, parents with lower educational attainment reported having more conversations with their kids regarding pertinent environmental knowledge and comprehension. Compared to their more educated peers, they also indicated a little higher level of confidence in their capacity to impart environmental information, foster desired environmental attitudes, and promote eco-friendly activities ([22]). Higher grades showed a marginally non-significant negative effect (coefficient = −0.051, *p* = 0.067) consistent with the previous studies (e.g., [12]), where younger students exhibit more pro-environmental behavior. This decline may be linked to a lack of exposure to environmental topics, as well as a lack of active learning approaches that could better foster pro-environmental behavior in students, and suggests the need for policy changes in curricula to ensure that environmental concepts are taught effectively across all grade levels ([25]). In conclusion, gender, grade, and parental education largely maintained their influence in the post-test, while health status showed no effects on the investigated domains, despite its baseline impact on attitudes. Similar to attitudes and perception, boys scored lower than girls in behavior, reinforcing the pattern of gender differences in pro-environmental engagement. This aligns with the existing research suggesting that girls and women are more likely to adopt pro-environmental behaviors than boys.

[25] ([25]), in their study of comparing differences between two groups (intervention and control) with respect to their environmentally responsible behavior, reported no statistically significant differences between groups. Similarly, the research conducted by [35] ([35]) found no difference in behavior scores in the intervention groups, thus revealing similar outcomes as our current study. The negative effects of the predictors on environmental protection and healthcare behavior in this study could stem from various factors. Pro-environmental behaviors and involvement in environmental education programs are positively correlated in certain studies ([8]), but the findings of other studies are conflicting or unclear ([49]). These inconsistencies may be attributed to factors such as program design, duration, delivery methods, and socio-cultural contexts, which have not been fully explored. Additionally, the mechanisms through which environmental education programs impact youth behavior remain under-researched. It is important to note that behavior is influenced by a complex set of factors beyond the scope of the program itself. For instance, individual attitudes, peer influence, and external environmental factors can also play significant roles. Previous studies highlight the dynamics between environmental education and youth behavior, highlighting the need for more holistic and context-sensitive approaches to these interventions ([11]). Thus, there is an urgent need for empirical research that uses rigorous methodological techniques and considers a variety of contextual elements to investigate the intricate relationship between environmental education programs and the formation of sustainable habits in young people.

Follow-up analysis—The follow-up test was incorporated to assess the sustainability of the intervention’s impact and to examine whether the acquired knowledge translates into lasting changes in behavior. Given the study’s premise that increased knowledge could drive action, the follow-up measure aimed to determine the long-term retention of knowledge and its potential to produce sustained shifts in students’ attitudes and behaviors. This additional measure helped evaluate whether the knowledge gained during the intervention persisted over time and whether it had enduring effects on environmental actions. Our follow-up analysis and comparison with the post-test revealed mixed outcomes across knowledge, attitudes, perceptions, and behaviors. Knowledge retention showed a slight decline in the intervention group over time, narrowing the gap between intervention and control. Attitudes improved significantly in the intervention group by follow-up, suggesting a delayed positive effect. Perception scores remained consistently higher in the intervention group, emphasizing the intervention’s sustained impact on perception. However, the intervention showed counterproductive effects on behavior at both post-test and follow-up. This aligns with findings from [11] ([11]), who also observed rebound effects that diminished pro-environmental behavior. [23] ([23]) found that their intervention increased participants’ self-reported pro-environmental behavior over time; however, the effect was independent of group assignment, indicating that study participation alone may have enhanced behavior. [43] ([43]) found that information alone is insufficient to drive pro-environmental behavior, especially when barriers are high or motivation is low. These results underscore the complexity of evaluating interventions across multiple domains. While the knowledge and perception gains are promising, they highlight the importance of developing strategies that not only enhance knowledge but also translate that knowledge into sustained behavioral practices. Addressing this gap is essential for achieving the long-term sustainability of positive behavior changes in future interventions. The fact that our intervention may not have been centered on developing students’ critical thinking and action competence, as suggested by [13] ([13]) in their explanation of “shaping human behavior”, could explain some of the findings regarding attitudes and behavior. Furthermore, a long-term, steady development of attitudes and skills in pro-environmental behavior could be seen as more successful than the delivery of this one-time environmental education intervention. Even when knowledge ratings have improved, we typically anticipate seeing a shift in people’s attitudes or behaviors or a rise in environmentally friendly actions. The notion that this association is far more complex and mediated by particular variables, values, beliefs, social relationships, and other external factors has put the simple linear model of increased knowledge leading to behavior change and that behavior can be taught into question ([24]). Our findings, which show no positive change in behavior, imply that in the future, we should concentrate on creating more complex models and include a variety of variables to clarify these relationships. The findings of this study offer valuable implications for educational policy and practice in Kosovo and similar low-resource urban contexts. The observed improvement in knowledge and perceptions among pupils following a short-term intervention demonstrates that even modest environmental education programs can raise awareness about critical issues such as air pollution. However, the limited influence on attitudes and the lack of behavioral change underscore the need for educational policy to go beyond knowledge transmission and foster a more immersive, sustained, and participatory learning approach. To effectively translate knowledge into behavior, environmental education should be embedded as a long-term, cross-disciplinary component of the national curriculum rather than a one-off or extracurricular initiative. This includes continuous teacher training, development of age-appropriate and culturally relevant materials, and incorporation of experiential learning (e.g., community projects, school-based air quality monitoring, or green school initiatives). Moreover, the significant role of parental education—particularly maternal education—highlights the importance of community and family engagement in school-based environmental interventions. Educational programs should consider a multi-stakeholder approach, involving parents through workshops, informational materials, and opportunities for joint learning activities, which may enhance the transfer of knowledge and environmental values at home. Finally, our results suggest a need to introduce environmental education early and reinforce it consistently through progressive, age-adapted content. Policies should ensure that environmental literacy is nurtured throughout the entire basic education cycle, with assessments of environmental competencies embedded in national education standards.

## 5. Limitations

Our study had a number of restrictions that might have impacted the outcomes. First, the intervention was followed up over a short period of 3 and 6 months only, limiting our ability to measure long-term outcomes. Second, no pre-tested and validated questionnaire existed before our study to measure knowledge, attitudes, and behaviors regarding air pollution in that specific setting. Therefore, we had to develop our own questionnaire and test it in the first pre-intervention round. As demonstrated in our first paper ([44]), some of the questions lacked internal consistency. Therefore, we had to adapt the questions and especially reduce the questions that made up the knowledge score. Third, motivating children to consistently participate in the program was challenging, which may have impacted the overall effectiveness of the intervention. While a dropout rate of 20% was accounted for in the power calculation, and indeed participating rate was sufficient to allow for reasonably precise estimates, untracked absenteeism due to sickness or other factors meant that we could not verify if all students in the intervention group received the full educational content. We assumed full participation, which may have introduced inconsistencies. The differences in participant composition may have influenced the results, as participants at different grade levels could have had varying capacities to engage with the intervention. While we attempted to control for these differences, future studies should consider matching participants more closely or using statistical methods to adjust for baseline differences. Unfortunately, because of privacy and confidentiality issues, we were not able to obtain the names of the respondents. Therefore, we could not follow each single child separately across the surveys but only made comparisons on the group levels. These factors should be taken into account when interpreting the findings, and further research is needed to better isolate the effects of the intervention. Fourth, our reliance on self-reported questionnaires may have led to biased responses, as children might have overestimated or underestimated their perceptions and behaviors. One could argue that with higher knowledge the children are more critical about their own behavior and attitudes and respond to these questions accordingly. These limitations highlight the need for longer-term studies and more robust tracking mechanisms in future research.

## 6. Conclusions

This study represents one of the first efforts in Kosovo to assess the impact of a structured school-based environmental education intervention on students’ knowledge, attitudes, perceptions, and behaviors related to air pollution. The findings indicate that while the intervention was successful in significantly improving students’ knowledge and perceptions, it had limited impact on changing attitudes and no measurable effect on pro-environmental behavior. These results reflect a broader challenge within environmental education—namely, that increasing awareness alone may not be sufficient to produce meaningful and sustained behavior change. Several limitations should be considered when interpreting these findings. The relatively short duration of the intervention and follow-up periods (3 and 6 months) limited the ability to observe long-term outcomes. The development and adaptation of a context-specific questionnaire, although necessary, may have influenced the reliability of the measures, particularly in the knowledge domain. Furthermore, challenges in maintaining consistent student participation—along with reliance on group-level rather than individual-level comparisons due to privacy constraints—may have introduced bias. The use of self-reported data also carries the risk of social desirability or response bias, particularly in younger age groups. Despite these limitations, the study provides valuable insights for future research and for educational policy and practice. The findings emphasize the importance of early and sustained environmental education within the school system, moving beyond one-time interventions toward integrated, long-term programming. The observed associations between maternal education and positive environmental attitudes, as well as gender differences in perception, suggest that environmental education efforts should not occur in isolation but rather in conjunction with family and community engagement strategies. For policymakers and educators in Kosovo and similar settings, this study underscores the need to embed environmental education more deeply into national curricula. Programs should be designed to address not only cognitive outcomes like knowledge and perception but also affective and behavioral dimensions. Approaches that incorporate participatory, experiential learning, intergenerational dialogue, and community-based action projects may be more effective in fostering the skills and values necessary for lasting environmental stewardship. Future research should focus on longitudinal studies that track individual-level changes over extended periods, employ validated measurement tools, and explore mechanisms that connect environmental knowledge to sustainable behavior. In this way, the educational sector can play a transformative role in cultivating environmentally responsible citizens and in contributing to broader efforts toward climate action and public health in Kosovo.

## Figures and Tables

**Figure 1 ejihpe-15-00069-f001:**
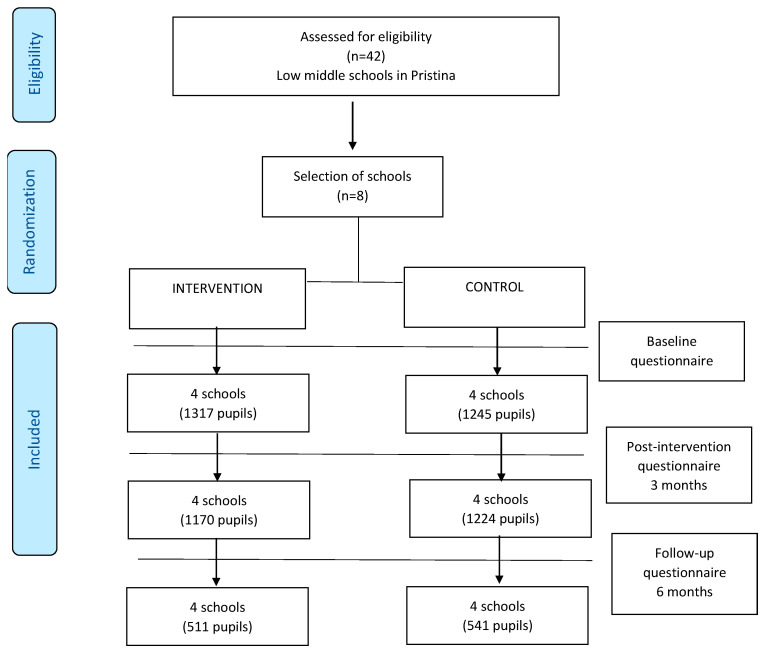
Flow diagram of the schools and participants of the study.

**Figure 2 ejihpe-15-00069-f002:**
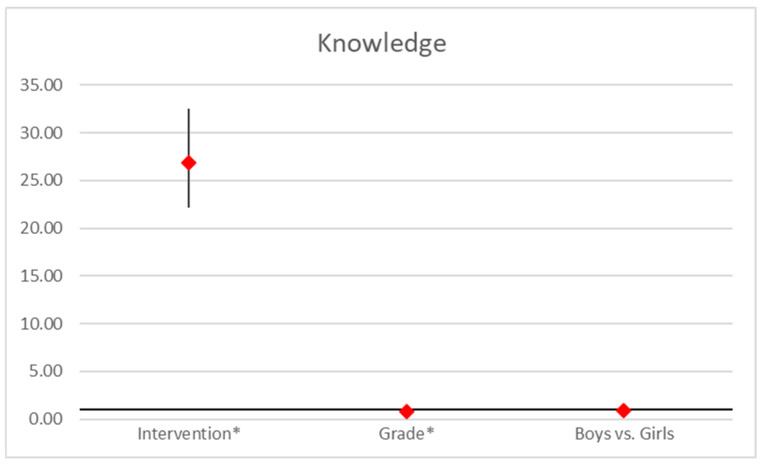
Regression analysis on knowledge scores. Point estimates and 95% confidence interval in odds ratios of ordered logistic regression analysis (*… *p* < 0.05).

**Figure 3 ejihpe-15-00069-f003:**
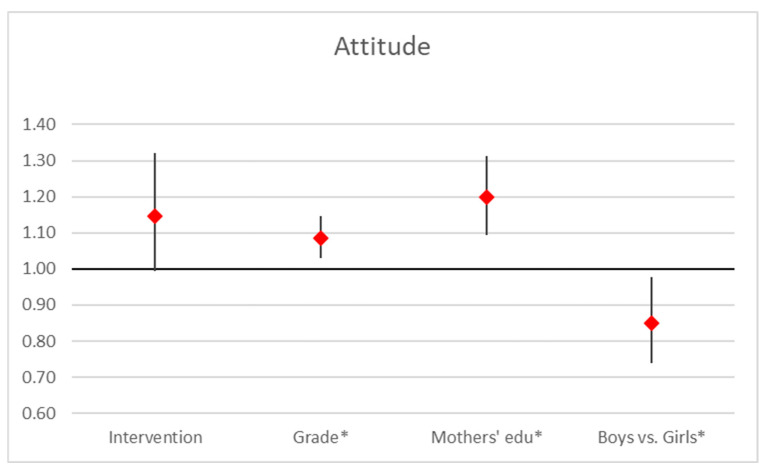
Regression analysis on attitude scores. Point estimates and 95% confidence interval in odds ratios of ordered logistic regression analysis (*… *p* < 0.05).

**Figure 4 ejihpe-15-00069-f004:**
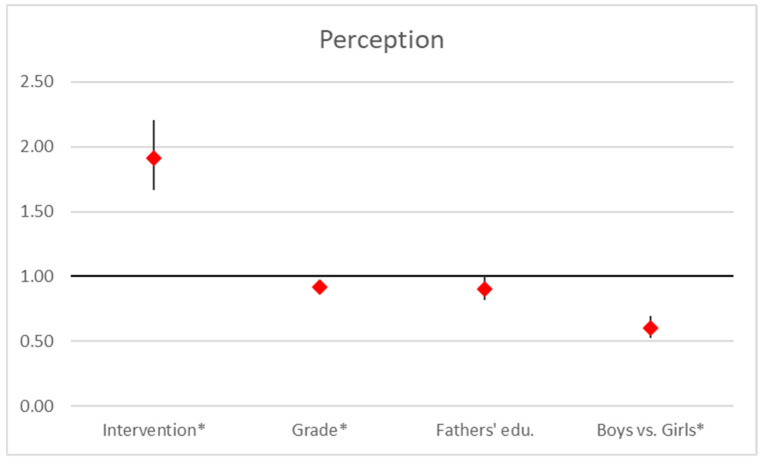
Regression analysis on perception scores. Point estimates and 95% confidence interval in odds ratios of ordered logistic regression analysis (*… *p* < 0.05).

**Figure 5 ejihpe-15-00069-f005:**
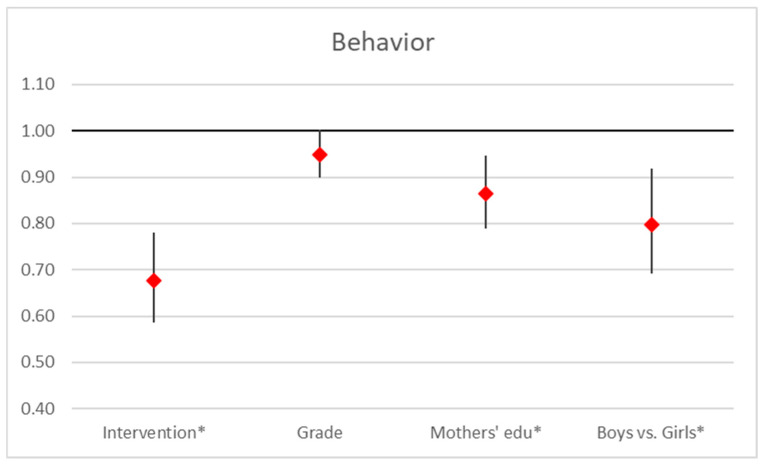
Regression analysis on behavior scores. Point estimates and 95% confidence interval in odds ratios of ordered logistic regression analysis (*… *p* < 0.05).

**Table 1 ejihpe-15-00069-t001:** Categorical variables in subjects assigned to intervention and control group at post-intervention and follow-up.

Variables N (%)	Post-Intervention		Follow-Up	
	Intervention (n = 1169)	Control (n = 1223)	Intervention (n = 510)	Control (n = 541)
Sex				
Males	533 (45.6%)	582 (47.6%)	246 (48.2%)	285 (52.6%)
Females	636 (54.4%)	641 (52.4%)	264 (51.8%)	256 (47.4%)
Grade				
5th Grade	80 (6.8%)	145 (11.8%)	72 (14.2%)	117 (21.6%)
6th Grade	260 (22.3%)	214 (17.5%)	159 (31.2%)	136 (25.3%)
7th Grade	258 (22.1%)	287 (23.5%)	163 (31.9%)	149 (27.5%)
8th Grade	275 (23.5%)	297 (24.3%)	101 (19.8%)	37 (6.8%)
9th Grade	296 (25.3%)	280 (22.9%)	15 (2.9)	102 (18.8%)
Father’s education				
No school	6 (0.5%)	14 (1.2%)	4 (0.8%)	4 (0.7%)
Primary school	133 (11.4%)	112 (9.2%)	66 (12.9%)	69 (12.8%)
Secondary school	349 (29.8%)	388 (31.7%)	120 (23.5%)	187 (34.6%)
University	681 (58.3%)	709 (57.9%)	320 (62.8%)	281 (51.9%)
Mothers’ education				
No school	45 (3.8%)	24 (1.9%)	19 (3.7%)	13 (2.4%)
Primary school	151 (12.9%)	85 (6.9%)	66 (12.9%)	42 (7.8%)
Secondary school	321 (27.5%)	389 (31.9%)	115 (22.6%)	200 (36.9%)
University	652 (55.8%)	725 (59.3%)	310 (60.8%)	286 (52.9%)
Health Status				
Poor	28 (2.4%)	18 (1.5%)	13 (2.6%)	9 (1.7%)
Fair	164 (14.1%)	138 (11.2%)	84 (16.5%)	60 (11.1%)
Good	475(40.6%)	491 (40.2%)	204 (40.0%)	226 (41.7%)
Very Good	502 (42.9%)	576 (47.1%)	209 (40.9%)	246 (45.5%)

**Table 2 ejihpe-15-00069-t002:** Comparison of intervention and control follow-up knowledge scores, with *t*-test.

Group	Number	Mean (Mean)	95% Confidence Interval	*p*-Value
Intervention	510	5.41 (0–8)	5.29; 5.54	
Controls	541	3.03 (0–7)	2.91; 3.15	
Both	1051	4.19 (0–8)	5.57; 5.89	
Difference		2.38	2.21; 2.55	<0.001

**Table 3 ejihpe-15-00069-t003:** Comparison of intervention and control follow-up attitude scores, with *t*-test.

Group	Number	Mean	95% Confidence Interval	*p*-Value
Intervention	510	8.18 (0–14)	7.93; 8.43	
Controls	541	7.72 (0–13)	7.52; 7.91	
Both	1051	7.94 (0–14)	7.78; 8.10	
Difference		0.46	0.15; 0.78	0.0038

**Table 4 ejihpe-15-00069-t004:** Comparison of intervention and control follow-up perception scores, with *t*-test.

Group	Number	Mean	95% Confidence Interval	*p*-Value
Intervention	510	22.69 (7–33)	22:32; 23:06	
Controls	541	21.37 (8–33)	21.03; 21.72	
Both	1051	22.01 (7–33)	21.76; 22.27	
Difference		1.32	0.81; 1.82	<0.001

**Table 5 ejihpe-15-00069-t005:** Comparison of intervention and control follow-up behavior scores, with *t*-test.

Group	Number	Mean	95% Confidence Interval	*p*-Value
Intervention	510	4.65 (0–9)	4.48, 4.81	
Controls	541	5.33 (0–9)	5.16, 5.50	
Both	1051	5.00 (0–9)	4.88, 5.12	
Difference		−0.68	−0.92; −0.45	<0.001

## Data Availability

Raw data from the surveys are available upon request from the corresponding author.

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
