# Peer review of "Effect of an Educational Intervention on Pupil’s Knowledge, Attitudes, Perceptions, and Behavior on Air Pollution in Public Schools in Pristina"

_ejihpe, 2025, doi:10.3390/ejihpe15050069_

Round 1
Reviewer 1 Report
Comments and Suggestions for Authors
The paper deals with an important environmental issue that deserves more attention in school education. The intervention carried out in schools and its evaluation are therefore highly relevant.
The following review contains more general comments. Further comments can be found in the manuscriptitself.
*** Comments on the introduction:
- The significance of the present study and the theoretical background are comprehensively presented in the introductory section.
- The research question at the end of the introductory section must be specified. The analysis does not only refer to the increase in knowledge, but also changes in attitude, perception and behavior are analyzed. Furthermore, the analysis is not only concerned with long-term changes, but also with short-term changes, i.e. both the data from the post-test and the follow-up test are analyzed. Through the various regressions that were calculated, further relations between the variables independent of the intervention are also discussed. This aspect is missing in the research questions.
*** Comments on the materials & methods section:
- The methods section contains too little information on Rasch analysis. No quality measures are presented, like the person-separation reliability. In addition, information should be provided about the rangethat was covered by person abilities and item difficulties (lowest and highest values). Maybe a Wright-mapcould be added to the appendix and/or a list of items including item difficulties. In addition, more detailed information on differential item analysis (DIF) would be helpful.
- It is not described how the scores for the different variables (knowledge/K, attitude/A, perception/P, behavior/B) were calculated. In addition, quality/reliability measures (e.g. Cronbach’s alpha) are missing for A, P and B.
- I miss a definition for attitudes and perceptions. When I look at the items in the appendix, it is not clear to me what exactly the difference is between these two variables. The behavior variable also needs to be defined, in that it only deals with self-reported behavior. This variable comprises two different types of behavior: Items 1 and 2 refer to environmental protection behavior and item 3 refers to protecting one's health. In my opinion, one would have to justify why these 2 types of behavior (which serve different purposes) can be grouped together in one category.
- It is unclear why just ordered logistic regressions were calculated. For the K-data a linear regressionwould have been appropriate, as the data are Rasch-transformed.
- It is unclear to me, why the data of the pre-test were not used for comparison purposes with those of the post-test (and/or follow up-test). If one wants to analyze the gain in knowledge caused by the intervention, then one should compare the results of the pre-test with those of the post-test. If one concentrates on the comparison of the control and intervention group in the post-test (or the follow-up test), then one must first show that there is no difference between the control and intervention group in the pre-test. This has not been reported so far and must be included.
- It is also unclear to me why regression analyses were calculated for the post-test and difference analyses (t-tests) for the follow-up test. This choice (which tests for which measurement time point) should be justified.
*** Comments on the results section:
- Table 1 shows differences between the intervention and control groups, e.g. in the subjects' grade level. These differences between the two groups then also change between the different measurement times. - The discussion should, therefore, address the extent to which these changes in the composition of the participants could also be responsible for differences between the two groups.
- The coefficient should be specified in the regression analysis. Why is the regression coefficient ß given, but not the odds ratios whereby the latter are better to interpret?
- What is the additional information given by the figures 2-5 in comparison to the description in the text? Should the significance level also be indicated by asterisks in the graphs? What is missing is the labeling of the Y-axis. Is it the regression coefficient ß?
- Regarding gender, it should be possible to indicate whether a higher proportion of boys or girls leads to a lower probability of environmentally conscious attitude, perception and behavior.
- It would be helpful for the tables 2-5 if the minimum and maximum values (range) could be added to the header (column „Mean“).
- I would appreciate if the results section could be supplemented with information on effect sizes.
*** Comments on the discussion:
- At the beginning of the discussion, the areas K, A, P and B are not clearly separated and seem to overlap.
- Perhaps it would be better to divide the discussion into sub-chapters to make the structure even more visible.
- I wonder a bit how the discussion fits in with the title and the original research questions. The focus there is on the effects of the intervention. However, the discussion is about many different factors influencing K, A, P and B, of which the intervention is only one. Is the aim, therefore, to compare the extent to which the intervention leads to stronger effects in the various target areas than other variables?
- In the discussion, other studies are cited as evidence to support the own results. However, reasoned explanations for the results are helpful too. In some cases, such possible explanations are missing. E.g., what could be the reasons why knowledge as well as attitudes, perceptions and behavior referring to air pollution decline with increasing age?
- It seems strange to me that all predictors (independent variables) have a negative effect on the criterion variable behavior. What are the reasons why the intervention leads to lower environmental protection behavior and/or health care behavior? The statement that behavior is also determined by other factors seems very general to me.
*** Comments on the appendix:
- The texts of the items should be provided in full length.
- The answer format should be included.
- For the K-items item difficulty and fit statistics (mean square values) could be added.
*** Please also consider the comments in the manuscript for revision.

Author Response
Comments provided in the document attached.

Reviewer 2 Report
Comments and Suggestions for Authors
Dear author(s),
I have read with much interest your paper titled “Effect of an Educational Intervention on air pollution among Public School Pupils in Pristina”.
The paper present useful data regarding the impact of an educational program implemented in schools in Pristina, Kosovo, in order to fill this gap and to assess the extent to which such initiatives can influence students' environmental knowledge, perceptions and behaviors.
Please find below few points which needs clarification / reanalysis / rewrites and/or additional information and suggestions for what could be done to improve the paper.
Title
To ensure a clearer and more comprehensive representation of the study, I recommend rewording the title. If you agree, some rewording alternatives could be: “Effect of an Educational Intervention on student’s knowledge, attitudes, perceptions, and behavior on air pollution among Public School in Pristina” / “Effect of an Educational Intervention on student’s awareness on air pollution among Public School in Pristina”.
Abstract
- For clarity and a better understanding of the approach and results I recommend to highlight a short background and minimum information on the research methodology.
Introduction
- In order to strengthen the theoretical foundation of the study, a clearer highlighting of the identified gap in the literature is required, explicitly outlining what existing gaps or limitations this research addresses and how it contributes to the expansion of knowledge in the field of environmental education and its impact on students' attitudes and behaviors
- Also, for a better understanding of the direction of the study, I recommend explicitly outlining the research questions in relation with the proposed aim.
Materials and methods
- I recommend restructuring the 2.1. methodology section (“Study design and data collection”), by highlighting in separate subsections: participants, research method and instruments (including measuring scale), data collection and analysis.
- Also, I recommend detailing of how training sessions are delivered in terms of: types of applied strategies and if these were identical for all levels 5-9th grade, periodicity and duration of sessions, number of the participants per series/group etc.
Results
- In the Results section, I recommend to provide in-depth interpretations of the results, explaining observed trends and offering concrete recommendations based on the research findings and literature data. (For example, how can it be explained that "the intervention has a significant negative effect on behavioral scores"? - line 197, How this could be addressed in future research and or training programs (interventions)?).
- Also, I recommend including the limitations of the research in a separate section.
Conclusions
- Also, in the conclusion section, it would be useful to emphasize how the results obtained respond to the research questions, thus strengthening the relevance and impact of the study.
Author Response

(The authors gave the same response as above.)

Reviewer 3 Report
Comments and Suggestions for Authors
Thank you for sharing the results of your study. I very much support any efforts to reduce exposure to air pollution and applaud the broader aims of the initiative. Further, I agree that younger age groups are rarely included in research studies relating to environmental awareness and behaviour. However, I am afraid that in its current form, I do not consider the paper to be ready for publication. I feel it is a under-developed, and would benefit from more information, and greater contextualisation in terms of both teh wider literature, and to the specific locale of Kosovo. Below I list some areas that I hope will help you in developing the paper further.
1) You rightly recognise that significant efforts, and research insights, have long been directed at raising community awareness of environmental issues on the proviso that increased knowledge will in turn drive changes in behaviour. I'm sure that you are also know that attempts to increase knowledge do not automatically lead to behaviour change: that is, the situation is much more complicated and nuanced. I thus wonder whether this paper is part of a larger study that, in addition to a questionnaire component, additionally conducted qualitative interviews (or similar) with individuals to determine the particular reasons for their limited behaviour change? If such interviews exist (or if there are other aspects to this study) I would very much advise that they be reported alongside the basic quantitative measures in order to add further nuance to the reasons for particular behaviour choices.
2) I would welcome justification for why maternal and paternal education were selected as variables in the study. Are these proxies for socio economic status? What other factors were considered as possibly affecting behaviours (for example, parental politics, or location of the child's house)?
3) I would welcome much more detail on the nature of the intervention and how it was developed to ensure maximum effect. That is, were the activities tested with target audiences? Did you consider working with children to determine the most effective way to communicate? If not, perhaps these limitations could be explicitly noted, and perhaps reflected upon for subsequent related research.
4) I would have liked more detail on the how constructs such as attitudes and behaviours were conceptualised and thereafter measured. There is a considerable research literature on learner attitudes which has not been cited.
5) The research design included a post intervention measure and a follow up measure. An explanation is needed here as to why a follow up knowledge test was required. (I am guessing because the study was based on the premise that knowledge leads to action and there was a desire to see how long knowledge lasts. However, if this premise is contested, then the need for a follow up knowledge test is also questionable).
6) Finally, given that the study was located in Kosovo, it would be appropriate to hear more about the particular context. Are there issues specific to Kosovo that would apply more to certain contexts elsewhere in the world, than others? Are findings particular to Pristina? And ultimately, what do the findings mean for policy makers and educators and parents living in the region (or in similar settings)? Currently, the conclusions listed were limited, non-specific and would benefit from more reflection.
I hope that these suggestions are helpful for developing this paper further. I also wish you and colleagues all the best in your efforts to ameliorate the dangers of local air pollution.
Author Response

(The authors gave the same response as above.)

Round 2
Reviewer 1 Report
Comments and Suggestions for Authors
Thank you for revising the manuscript, which has become much clearer for me now. Nevertheless, there are a few minor comments
Words with inappropriate hyphens appear repeatedly in the text. These hyphens should be deleted.
Line 113-115: This sentence merely repeats the content of the previous sentence and can be deleted.
Line 117 and 325: The term ‘awareness’ is used here. Is it synonymous with ‘knowledge’ or ‘perception’? A consistent use of terms would be better.
Line 119: Is ‘attitude’ missing here alongside changes in ‘knowledge’ and ‘behavior’?
Line 203: The knowledge questions in the appendix are not only closed questions (true/false), but also open questions – an aspect that should be mentioned in the text. Question 10 even allows several answers. How was this question converted into a 0 - 1 code? This should also be mentioned in the text
Line 228: “and were advised” -> “and students were advised”
Line 255, 265 and 382: Please add in what way there were differences between genders. Although this is now mentioned in the conclusion, it should already be included in the results section and in the discussion.
Line 286: The values “7.55” and “4.01” do not match those in Table 2.
Table 2-5: Please add effect sizes for the differences between the intervention and the control group.
Line 332: Please delete the word “resulted” (there are 2 verbs in a row).
Line 337: “Our findings show a consistent effect of grade level on attitude scores, with a decline in positive attitudes as grade level increases.” This statement is not consistent with Figure 3. In Figure 3, the attitude increases with grade level.
Line 392: “Saracli et al., 2014 found that students with university-educated mothers showed greater environmental responsibility and pro-environmental behaviors than those whose mothers had only elementary education [47]”. This literature does not match the results, because Figure 5 shows that a higher education of mothers leads to a lower environmental behavior of their children. The result must therefore be explained differently than with the study by Saracli et al. (I still wonder whether an error may have occurred in the evaluation of the behavior, as all the factors examined have a negative effect on behavior.)
Line 404: Here it must be stated how the two groups in Krnel & Naglic's study differed with regard to the independent variable: Did one group receive an instructional intervention and the other did not? As the sentence is now, it is a circular argument.
Line 490: “addressing the research question related to knowledge improvement” - I would delete this part of the sentence, as the research question refers to changes in knowledge, attitude, perception and behavior.
Referring to my comment no. 5 in the first review round: (i) I would appreciate it if the distinction between attitude and perception could be added to the manuscript in a footnote. (ii) It would also be sensible to explain in a footnote why the two types of behavior (environment- and health-related) were combined. The text of the authors’ response can be used for this purpose.
Referring to my comment no. 7 in the first review round: Here too, the text of the authors’ response could be used as a footnote to explain why a direct comparison of the pretest and posttest data was not possible.
Author Response
Cover letter with answers provided in the word document

Reviewer 2 Report
Comments and Suggestions for Authors
Dear author(s),
I have read with much interest your revised paper renamed “Effect of an Educational Intervention on student’s knowledge, attitudes, perceptions, and behavior on air pollution among Public School in Pristina”.
The manuscript has been improved regarding the most aspects addressed in the review report.
Thank you for your efforts to bring these improvements to the revised form.
I have the following two minor suggestions:
- I recommend including in the Abstract section information regarding the research method and instrument as well as the number of participants involved.
- I maintain the recommendation of explicitly highlighting the research question(s) in relation to the research aim. This reinforces the scientific rigor of the study and may facilitate readers' understanding of how the results responds to the research problem addressed in the study.
Author Response

(The authors gave the same response as above.)

Reviewer 3 Report
Comments and Suggestions for Authors
I acknowledge that effort has been made to seemingly address specific suggestions from the three reviewers.
However, I have to say that my own suggestions were made to support the re-development of the paper in full, not as specific instances to be addressed prior to acceptance. That is, I consider the reporting of quantitative data is only useful if complemented with some qualitative data seeking to unpack the nature of responses, and some reflection about the particular pedagogical nature of the intervention. Thus, I was hoping that you would re-frame this quantitative data set in a new, more wide-reaching, paper.
More practically, any feedback to my own comments has been very difficult to read. The pop up box does not allow me to read the full response. This may be a system problem, but I was surprised that more detailed explanations were not provided in a feedback letter. With respect to other comments, the feedback simply says 'addressed in manuscript', but it not particularly clear where (or whether any text was removed).
Relatedly, I could not see that my comment (which you have numbered 6) about 'what could these findings mean' for policy makers and educators and parents in the region has been addressed at all. Offering a more substantive commentary here is, in my eyes, key to 'lifting' this paper from a report of data to a more reflective discussion that would potentially offer a novel contribution to scholarship
Further, I cannot access the feedback from reviewers 1 and 2 - which would have been useful - so cannot really determine whether these points have been addressed or not. Again, this may be a system issue (and not yours), but it has meant my re-review of your work has been compromised.
Author Response

(The authors gave the same response as above.)

Round 3
Reviewer 3 Report
Comments and Suggestions for Authors
NA
Author Response
We thank the reviewer for his/her supportive comments